# Risk Evaluation and Attack Detection in Heterogeneous IoMT Devices Using Hybrid Fuzzy Logic Analytical Approach

**DOI:** 10.3390/s24103223

**Published:** 2024-05-19

**Authors:** Bharanidharan Shanmugam, Sami Azam

**Affiliations:** Energy and Resource Institute, Faculty of Science and Technology, Charles Darwin University, Darwin, NT 0810, Australiasami.azam@cdu.edu.au (S.A.)

**Keywords:** analytical hierarchy process, IoMT, risk assessment, sniffing, jamming, injection

## Abstract

The rapidly expanding Internet of Medical Things (IoMT) landscape fosters enormous opportunities for personalized healthcare, yet it also exposes patients and healthcare systems to diverse security threats. Heterogeneous IoMT devices present challenges that need comprehensive risk assessment due to their varying functionality, protocols, and vulnerabilities. Hence, to achieve the goal of having risk-free IoMT devices, the authors used a hybrid approach using fuzzy logic and the Fuzzy Analytical Hierarchy Process (FAHP) to evaluate risks, providing effective and useful results for developers and researchers. The presented approach specifies qualitative descriptors such as the frequency of occurrence, consequence severity, weight factor, and risk level. A case study with risk events in three different IoMT devices was carried out to illustrate the proposed method. We performed a Bluetooth Low Energy (BLE) attack on an oximeter, smartwatch, and smart peak flow meter to discover their vulnerabilities. Using the FAHP method, we calculated fuzzy weights and risk levels, which helped us to prioritize criteria and alternatives in decision-making. Smartwatches were found to have a risk level of 8.57 for injection attacks, which is of extreme importance and needs immediate attention. Conversely, jamming attacks registered the lowest risk level of 1, with 9 being the maximum risk level and 1 the minimum. Based on this risk assessment, appropriate security measures can be implemented to address the severity of potential threats. The findings will assist healthcare industry decision-makers in evaluating the relative importance of risk factors, aiding informed decisions through weight comparison.

## 1. Introduction

Traditional medical scenarios involve healthcare professionals manually collecting and managing the health data of patients using medical equipment such as stethoscopes, thermometers, and blood pressure monitors. The health data are usually recorded on paper or in the form of electronic health records (EHRs). Contrary to this, medical IoT scenarios use connected devices, such as wearables and implantable devices, to collect and transmit real-time patient health information [1].

With the emergence of the Internet of Things (IoT) in healthcare, a huge number of devices need to be connected to the Internet, and such a system is referred to as the Internet of Medical Things (IoMT) [2]. The IoMT is a network of medical devices connected to the cloud for sending and receiving health data [3]. These devices generate massive data, which need careful monitoring. Hence, to keep risks under control, continuous risk assessment and management are becoming increasingly important [4]. According to Data Bridge Market Research [5], it is estimated that the IoMT market will surge from USD 48.69 billion in 2021 to USD 270.4 billion in 2029 [6].

There is the possibility that a single flaw could cripple vital health infrastructure [7]. It is therefore crucial to perform a risk assessment in order to achieve a risk-free IoMT device. This is evidenced by the fact that healthcare needs are expected to rise as the population ages. Even though the IoMT contributes to rapidly growing needs, it is also highly vulnerable to cyber-attacks that pose various threats targeting sensitive health data and systems [8].

Risk assessment helps to address all these security concerns associated with IoMT devices. It involves evaluating potential risks, vulnerabilities, and their impact on the security and privacy of medical IoT devices. The overall goal is to identify, analyse, and prioritize possible risks to develop and implement effective mitigation strategies. This paper introduces a risk assessment framework, which extends our previous work [9]. We have introduced a hybrid risk assessment (HRA) approach involving fuzzy logic and the Fuzzy Analytic Hierarchy Process (FAHP) for the risk assessment of heterogeneous IoMT devices.

Lotfi Zadeh [10] originally introduced fuzzy logic in 1965 as an improved form of Boolean logic based on mathematical fuzzy sets. In real-life problems, fuzzy logic can be crucial, especially when we cannot determine whether a given solution is correct or incorrect. By being similar to human thought, it resolves the ambiguity and inaccuracy that may arise when making decisions [11]. On the other hand, the Fuzzy AHP is an effective and useful method that provides crisp and valuable results in a pair-wise matrix [12]. In the AHP, the complex problem is always broken down into small problems and arranged hierarchically. Each level of the hierarchy represents a different set of criteria, sub-criteria, or alternatives [13].

### 1.1. Fuzzy Logic

A fuzzy logic approach is based on mathematical principles to represent knowledge in terms of degrees of membership and truth. It reflects the thinking skills and intellectual abilities of people in devising approaches and different circumstances [14]. Using fuzzy logic in risk assessment has been a successful strategy for dealing with risks, and it works efficiently with hybrid data. It can handle the ambiguity and uncertainty inherent in many risk assessments using linguistic variables and fuzzy sets [15].

Fuzzy logic has the advantage of modelling a complex problem using linguistic variables to express specific logic rules. A fuzzy inference system consists of three processes: fuzzification, inference engine, and defuzzification. Membership functions are defined as input variables, which are applied to their actual values during fuzzification. As part of the inference process, the truth value for the foundation of each rule is computed, which will then be incorporated into the concluding part. These sets of rules are generated with the IF–THEN statement. In the defuzzification process, a fuzzy quantity is converted to a precise value.

### 1.2. FAHP

The Analytical Hierarchy Process (AHP) is a widely used technique to handle problems with multiple conflicting criteria. It provides a valid decision-making process based on hierarchical reasoning and a pair-wise comparison of the criteria [16].

By using the AHP, we can reduce the bias associated with multiple-criteria decision-making (MCDM). An extension of fuzzy logic with the AHP called the FAHP can overcome challenges associated with subjectivity and uncertainty. Considering the imprecise and uncertain nature of human decision-making, the FAHP is often used to address problems associated with MCDM [17].

### 1.3. Contributions

As a contribution, this paper aims to fulfil the following three objectives:First is the adaptation of fuzzy logic and the Fuzzy Analytic Hierarchy Process (FAHP) in the context of everyday IoMT devices.Second, our research seeks to understand the causes of risk, raise risk awareness, and assist engineers and/or operators in determining which risk should be taken into account first. Our hybrid risk assessment process enables an accurate representation of the levels and risk scores with respect to risk events.Third, we have performed attacks on three different IoMT devices to prove the vulnerabilities during the pairing process.

### 1.4. Organization of Paper

In Section 2, we discuss the literature review, including fuzzy logic, the FAHP, and the hybrid method for the risk assessment of medical devices. Section 3 discusses the hybrid risk assessment process (HRA), which is the main contribution of this paper, where we utilize the HRA process with membership functions and the FAHP. Thereafter, a case study on the risk assessment of three IoMT devices is presented in Section 4 to demonstrate the application of the proposed risk assessment process. It presents attack scenarios along with vulnerabilities. Finally, Section 5 gives the conclusion and a summary of the preliminary benefits of using the proposed methodology in risk analysis, followed by future work that could expand in the next paper. In Appendix A, we include detailed calculations for weights and risk levels.

## 2. Literature Review

To summarize and collect research studies, we conducted a detailed analysis of the existing research pursuits in medical devices. Our comprehensive analysis encompasses a literature review pertaining to the IoMT, as outlined in our previous paper [9]. In the current paper, we review the literature encompassing topics such as fuzzy logic, the Fuzzy AHP, and the hybrid approach combining these two methods.

Fuzzy multi-criteria decision-making is widely used with incomplete or imprecise data, such as in [18], and a fuzzy set is used as an alternative to conventional decision-making. The goal of this paper is to summarize different types of fuzzy MCDM approaches with respect to their areas.

A hybrid MCDM framework was proposed in [19], which includes the AHP and Technique for Order Preference by Similarity to Ideal Solution (TOPSIS), where the weights of attributes are derived by the AHP method, and a security assessment is performed based on the TOPSIS method. Using the proposed framework, future guidelines can be formulated for selecting the best security solutions for IoMT-based systems, which can then be used to develop more frameworks. A further study is required to extend the framework by including more security requirements.

To identify security risks in medical devices, a Fuzzy AHP TOPSIS method was developed in [20], allowing manufacturers to take security into account from the beginning of the design process. The study proposed a security assessment of various medical devices and investigated a conceptual model that includes the increased integration of security principles into the design and implementation of medical instruments, as well as data protection during handling. The suggested framework has the capability of checking the security of different medical devices and can also enhance interoperability.

An overview of the present healthcare situation is presented in [21] using a layered approach. The paper also evaluates security breaches in healthcare through a hybrid fuzzy-based methodology, AHP-TOPSIS. However, due to the large scope of healthcare, the research only focuses on basic information security scenarios. The approach presented in [22] for mobile health applications was developed by adopting AHP and fuzzy TOPSIS, which is further discussed through a numerical case example. The AHP method was used to determine the weights of criteria and sub-criteria, and the fuzzy-TOPSIS method was used to determine the final ranking of the application. However, future research is recommended due to the limitations of both methods. A fuzzy inference system (FIS) was designed and applied to develop a risk assessment process in [23]. The study shows that the developed approach could be applied as a practical model for evaluating occupational health risks. The weight for each risk criterion is used to calculate the risk level by using a fuzzy approach. The above-mentioned studies show that none of them used the Hybrid FAHP method for the risk assessment of heterogeneous IoMT devices. Thus, in our study, we used the hybrid risk assessment approach to overcome uncertainty challenges.

## 3. Hybrid Risk Assessment Process

Assessing the degree of risk in heterogeneous IoMT devices is more challenging when considering the general problem of interpreting the unconstrained behaviour of these devices. A detailed systematic literature review was conducted for the risk assessment, and the methodology used by the authors was also determined. To address heterogeneity and security, in this study, we selected a hybrid approach, which is the Fuzzy AHP methodology. It has been proven by various researchers that the Hybrid AHP is better for providing informed decisions along with their weights. A flowchart for hybrid risk assessment is presented in Figure 1, which describes the combined process of fuzzy logic and the AHP.

### 3.1. Applications of Fuzzy Logic

Fuzzy logic allows the modelling of uncertain information by using fuzzy sets to represent concepts that have a degree of membership in a set, rather than being a true or false value. Three risk parameters are used to assess the overall risk level of IoMT devices, which are the frequency of occurrence (FO), the severity of consequences (SC), and the risk level (RL).

Fuzzy logic provides calculated risk scores and levels according to occurrence and consequences based on a membership function (MF). There are various forms of MFs: trapezoidal, triangular, Gaussian, bell-shaped, etc.

In our paper, selecting an appropriate membership function (MF) holds paramount importance for ensuring an accurate and efficient evaluation. As emphasized by the authors in [24,25], the primary requirement for an MF is its ability to range between 0 and 1. Among the vast array of options, triangular MFs have captivated our attention for their inherent simplicity and efficiency in handling uncertainty. Defined by just three parameters, they offer an intuitive framework, facilitating transparent risk assessment communication. Furthermore, their streamlined nature expedites computational processes, ensuring that we can navigate vast data sets with agility and precision. Our chosen IoMT devices involve factors like the severity of potential attacks and the likelihood of their occurrence. Triangular MFs will help capture the gradual nature of risk factors in IoMT security, allowing for seamless transitions between linguistic terms. They also enhance the interpretation of risk assessment outcomes, thereby increasing its reliability and precision for evaluating security-related scenarios.

To understand in more detail, we use a triangular MF for both occurrence and consequences, and the mathematical formulation is presented in Figure 2. Equation (1) is used to formulate the fuzzy triangular MF, and for notation, we use *l*, *m*, and *u*.
(1)$$\left\{\begin{array}{c} \;0,\;x<l,\;\\ \;(x-l)/\left(m-l\right),\;l\leq x\;\leq m,\;\\ \;\left(u-x\right)/\left(u-m\right),\;m\leq x\leq u,\;\\ \;0,\;x>u.\; \end{array}\right.$$

A triangular MF is defined by three parameters: a left base, a peak, and a right base. Here, *x* is the input value, X is the degree of membership, and *l*, *m*, and *u* are the three parameters, which denote the smallest possible value, the most promising value, and the largest possible value. The degree of membership is a number between 0 and 1, which represents how well the input value matches the fuzzy set. The value of *x* is shown on the horizontal axis, and the degree of membership is shown on the vertical axis. The fuzzy value is represented as $\mu_{\tilde{A}}\left(x\right)$ = (*l*, *m*, *u*), where these three numbers together are known as fuzzy numbers, which are associated with the membership function. The three numbers are the lower, middle, and upper ends of the triangle on the *x*-axis. Assigning a single number to any term is not justified, as we may have decimal values in between two numbers.

#### 3.1.1. Risk Identification

Risk identification is performed in two steps. In the first step, fuzzy risk analysis is carried out from the prospective harmful event level to the group level. In the second step, FAHP information is aggregated at the group level in order to obtain an overall risk level for the risk assessment of the IoMT devices. Here, in our study, three risks are identified based on the literature study. After identifying risks, each risk is evaluated based on three qualitative descriptors: FO, SC and RL.

#### 3.1.2. Fuzzification

The second step here is fuzzification, which is the interface between the input and the fuzzy inference engine [27]. It converts inputs into fuzzy qualitative descriptors and determines the degree to which each fuzzy set belongs, facilitating decision-making. During the process, the membership functions are defined as the input variables, which are applied to their actual values to determine the degree of membership for each rule [14].

#### 3.1.3. Fuzzy Inference Engine (FIS)

Once the values are converted, the next step is fuzzy inference, which is the process of translating the equivalent of the input data into the rule base. It is the actual brain of the fuzzy logic control system and defines the MF for each parameter [28]. The value of the membership function derives the outcome of the system [29]. The fuzzy inference system window “FIS Editor” is used in MATLAB for this process in the study. The fuzzy method is designed to acquire the risk value.

In our paper, the goal is to create MFs that reflect how security experts understand risk factors in IoMT devices. For both FO and CS, the values use equal intervals, which creates a relatively uniform distribution of membership degrees across the scale (0–5). The parameters align with the intuitive notion of severity and consequences, becoming more severe as the membership degree moves from low to high and very high. These triangles slightly overlap, allowing for values to have partial membership in two categories, which reflects the uncertainty in assigning a specific value, as clearly shown in Table 1 and Table 2.

In this analysis, rule bases were created from the input data with Mamdani, and the values in the risk analysis were calculated one by one. In the triangular membership function for occurrence values, low [0 1.25 2.5], medium [1 2.25 3.5] and high [2.5 3.75 5] value parameters were assigned. Figure 3 shows the membership functions of the frequency of occurrence shown in Table 1. Each qualitative descriptor of FO has a range to describe, and a mid-point of the estimated frequency is used in each category to obtain an approximate numerical value. For example, the qualitative expression “Low” is defined to cover the range of FO between 0 and 2.5, and their approximate numerical value is 1.25. The difference between 0 and 1.25 is equivalent to the difference between 1.25 and 2.5.

The consequence values are also in the triangular membership function; for negligible consequences, the value is [0 0.75 1.5]; for mild consequences, the value is [0.5 1.25 2]; for medium consequences, the value is [1.5 2.25 3]; for high consequences, the value is [2.5 3.25 4]; and for very high consequences, the value parameter [3.5 4.25 5] is assigned. Figure 4 shows the window of the MATLAB software program (r2023b) with the consequence values shown in Table 2. In this study, three and five qualitative expressions are used to describe FO and SC, respectively, but this is not necessary. There is flexibility in these descriptors, and they depend on the particular case.

Risk can be described by the degree to which it belongs to the qualitative expressions “very low”, “low”, “medium”, “high”, and “very high”, which are referred to as risk expressions. Table 3 shows the qualitative descriptor categories of risk level. A Gaussian MF is used for RL representation, as shown in Figure 5.

### 3.2. Development of Fuzzy Rule Base

The step following decisions about fuzzy membership functions is to understand the inference process to draw a conclusion from a set of fuzzy rules. Fuzzy rules can be achieved using various approaches, including expert opinion, data collection, and engineering knowledge, but they are not mutually exclusive, and a combination is usually the most effective approach. In the fuzzy rule base, fuzzy rules are determined by the number of qualitative descriptors rather than numerical values, making them a natural platform to deliver expert judgements and engineering knowledge [30]. The fuzzy rule base should cover all matches between inputs and outputs, and the rules should be chosen in a way that minimizes the possibility of contradictions and unwanted interactions between the rules.

The number of fuzzy rules in the fuzzy rule base depends on the number of qualitative descriptors used to represent the frequency of occurrence and the severity of consequences. It consists of a set of fuzzy if–then rules and is the core of a fuzzy logic system. For example, in the present study, there are three qualitative descriptors for the frequency of occurrence and five for the severity of consequences; the fuzzy rule base, therefore, consists of (3 × 5) = 15 fuzzy rules, which are listed in Table 4.

### 3.3. FAHP Analysis

The evaluated risk score of the events will feed the FAHP. It is a process for assessing the significance of a range of processes in a problem in order to solve complicated decision-making problems. The FAHP also has the advantage of being easy to integrate with a wide variety of techniques. It is mostly used in crisp information judgement implementation and is widely used for tackling multi-criteria decision-making problems in real situations [31]. To determine the relative contribution of each risk factor to the overall risk level, the weight must be considered so that the risk assessment can progress. The process starts with pair-wise comparison to derive the relative importance of the risk factors. Table 5 presents the risk level representation for the FAHP.

#### 3.3.1. Pair-Wise Comparison Matrix

A pair-wise comparison matrix is created with the help of the scale of relative importance, which determines the relative importance of different attributes or criteria with respect to the goal [17]. Based on an estimation scheme, each qualitative descriptor is paired with a triangular fuzzy number, which is then used to convert experts’ judgements into a comparison matrix. The arithmetic operations for two triangular fuzzy numbers, ${\tilde{a}}_{p}(t_{p}^{l}$, $t_{p}^{m},t_{p}^{u}$) and ${\tilde{a}}_{q}(t_{q}^{l}$, $t_{q}^{m},t_{q}^{u}$), are considered in the construction of the fuzzy pair-wise comparison matrix, where $t_{p}^{l}$, $t_{p}^{m},t_{p}^{u}$ are numbers from 0 to 5 with the relationship $t_{p}^{l}\leq$ $t_{p}^{m}\leq t_{p}^{u}.$ $t_{p}^{l}$ and $t_{p}^{u}$ correspond to the lower and upper values of a range to describe the *p*th qualitative descriptor. $t_{p}^{m}$ refers to the most likely value to represent the *p*th qualitative descriptor.

The arithmetic operations on fuzzy numbers are defined as follows:(2)$${\tilde{a}}_{p}⊗{\tilde{a}}_{q}=\left(t_{p}^{l}\times t_{q}^{l},t_{p}^{m}\times t_{q}^{m},t_{p}^{u}\times t_{q}^{u}\right)$$

Here, ⊗ represents fuzzy logic multiplication. If there are *m* experts in the risk assessment group, the element in a fuzzy pair-wise comparison matrix can be calculated using the formula below:(3)$${\tilde{a}}_{i,j}=\left(\frac{1}{m}\right)⊗\left(e_{i,j}^{1}⊕e_{i,j}^{2}⊕\ldots e_{i,j}^{k}\ldots ⊕e_{i,j}^{m}\right)\;{\tilde{a}}_{j,i}=\frac{1}{{\tilde{a}}_{i,j}}$$
where ${\tilde{a}}_{i,j}$ is the relative importance of event *i* compared with event *j*, and $e_{i,j}^{k}$ stands for the kth expert judgement in the triangular fuzzy number format.

Using Equation (4), we construct the pair-wise comparison matrix, which has comparisons in pairs, and select the appropriate linguistic value. An *n* x *n* pair-wise comparison matrix can be obtained as follows:(4)$$\tilde{A}=\left(\begin{array}{c} \begin{array}{cc} {\tilde{a}}_{1,1} & {\tilde{a}}_{1,2}\\ {\tilde{a}}_{2,1} & {\tilde{a}}_{2,2} \end{array}\;\begin{array}{cc} \cdots & {\tilde{a}}_{1,n}\\ \cdots & {\tilde{a}}_{2,n} \end{array}\\ \begin{array}{cc} \vdots & \vdots \\ {\tilde{a}}_{n,1} & {\tilde{a}}_{n,2} \end{array}\begin{array}{cc} \ddots {\tilde{a}}_{i,j} & \vdots \\ \cdots & {\tilde{a}}_{n,n} \end{array} \end{array}\right)$$

Here, $\tilde{A}$ is the comparison matrix, and n represents the number of criteria or alternatives. The value $a_{ji}$ demonstrates the relative significance of criteria i ($c_{i}$) in comparison with criteria j ($c_{j}$) on Saaty’s scale. The matrix represents the reciprocal relationships between the elements above and below the diagonal of the matrix. The diagonal entries of the matrix are all 1, as a criterion is always equally important to itself. The entries above the diagonal indicate the relative importance of the criterion in the row compared to the one in the column. The entries below the diagonal are the reciprocals of the entries above the diagonal. The fuzzy numbers in the matrix represent the uncertainty inherent in the comparisons. The fuzzy pair-wise comparison matrix is used to compute the weights of the criteria or alternatives. These weights represent the relative importance of each criterion or alternative in the overall decision.

#### 3.3.2. Weight Factor Calculation

Weight factors (WFs) can be calculated using the geometric mean method:(5)$$\tilde{f_{i}}={\left({\tilde{a}}_{i,1}⊗{\tilde{a}}_{i,2}⊗\ldots {\tilde{a}}_{i,j}\ldots ⊗{\tilde{a}}_{i,n}\right)}^{1/n}\;{=(\left(t_{i,1}^{l}\times t_{i,2}^{l}\times \ldots t_{i,j}^{l}\ldots \times t_{i,n}^{l}\right)}^{\frac{1}{n}},\;{\left(t_{i,1}^{m}\times t_{i,2}^{m}\times \ldots t_{i,j}^{m}\ldots \times t_{i,n}^{m}\right)}^{\frac{1}{n}}\;{\left(t_{i,1}^{u}\times t_{i,2}^{u}\times \ldots t_{i,j}^{u}\ldots \times t_{i,n}^{u}\right)}^{\frac{1}{n}})$$
(6)$${\tilde{w}}_{i}=\frac{\tilde{f_{i}}}{\tilde{f_{1}}⊕\tilde{f_{2}}⊕\ldots \tilde{f_{j}}\ldots ⊕\tilde{f_{n}}}$$

Here, fⅈ~ is the geometric mean of the ith row in the fuzzy pair-wise comparison matrix, and w~i is the fuzzy WF of the ith event.

#### 3.3.3. Defuzzification

Since the outputs of geometric mean methods are triangular fuzzy WFs, a defuzzification approach is employed to convert a triangular fuzzy WF to the corresponding WF, where the FAHP employs the proposed defuzzification approach [32]. The defuzzification method used here is the Centre of Area (COA), the formula for which is given in Equation (7):(7)DFwⅈ~=til+tim+tiu3
where DFwⅈ~ is the defuzzified mean value of the fuzzy WF. wi can be calculated as follows:(8)wi=DFwⅈ~ΣDFwⅈ~

Based on the risk levels (RLs) of the factors and their corresponding WFs, the overall risk level of heterogeneous IoMT devices can be calculated as in Equation (9), where $RL_{i}$ is the RL of the *i*th risk category, $w_{i}$ stands for the weight factor of the *i*th risk category, and RL is the overall risk level of an IoMT device.
(9)$$\mathrm{RL}=\sum_{i=1}^{n}RL_{i}w_{i}$$

On the basis of the RLs of risk events and the corresponding WFs obtained, the overall RL for IoMT devices discussed in the case study can be calculated.

To summarize, we have discussed the overall methodology, which focuses on determining the relative importance of assessing the risk levels of heterogeneous IoMT devices. A pair-wise comparison matrix is constructed using triangular fuzzy numbers, representing the uncertainty inherent in comparisons. The weight factor for each criterion is computed using the geometric mean method, which includes the calculation of the geometric mean of each row in the fuzzy pair-wise comparison matrix to obtain fuzzy WFs. These fuzzy WFs are then defuzzified using the CoA approach to convert them into corresponding WF values. Finally, the overall RL of heterogeneous IoMT devices is determined by combining the RLs of each factor with their corresponding WFs.

## 4. Case Study

To validate the proposed approach, a case study on attacks on three IoMT devices was performed. We considered three risk events or attacks named sniffing, jamming, and injection attacks on an oximeter, smartwatch, and smart peak flow meter. The oximeter is sourced from Xuzhou Yongkang Electronic Science Technology Co., Ltd., Xuzhou, China, and the application used is AiLink. The smartwatch is procured from Wgzixezn, Xuzhou, China, and the application used is DeepFit. The smart peak flow meter is bought from the Chongqing Moffy Innovation Technology Co., Ltd. of Chongqing City, China, and uses the Sonmol PEF application. 

An overview of the three devices is presented in Figure 6. Selecting these devices for risk assessment involves considering their widespread usage, their criticality in healthcare, potential risks associated with their operation, and the impact of those risks on patient safety. In addition, more research needs to be conducted on the risk assessment of these devices.

### 4.1. Selected Devices for Testing

An oximeter is a handheld clip device used to measure oxygen saturation. It is portable, making it easy to use at rest and during exercise [35]. While oximeters are generally user-friendly, they rely on accurate sensor readings and proper calibration to provide reliable data. They can be used in critical care settings, like emergency rooms, clinics, and hospitals, to monitor patients with respiratory conditions or at home. Saturation levels of oxygen are vital indicators of respiratory function and oxygen delivery to tissues and, if not addressed promptly, can lead to severe complications, including organ damage or failure [36]. Oximeters are subject to regulatory standards and guidelines to ensure their safety and effectiveness. By conducting a risk assessment on oximeters, compliance gaps can be identified and improved to ensure adherence to regulatory requirements, such as FDA guidelines for medical device cybersecurity.

The smart peak flow meter has been designed to monitor lung function and assess the severity of airway obstruction [34]. It connects wirelessly to smartphones and is used by asthma patients to monitor their clinical progress, track trends, and provide alerts, which can facilitate early intervention and prevent serious complications. Regular monitoring helps patients and healthcare providers manage these chronic conditions effectively.

The next device does not require an introduction, as it has become widely popular among consumers due to its multifunctionality, including features for tracking health, such as heart rate monitoring, activity tracking, sleep analysis, and even electrocardiography. With smartwatches, users have the convenience of real-time access to their physiological parameters throughout the day. Despite their benefits, smartwatches also pose risks due to their privacy, security, and accuracy problems. Data breaches, unauthorized access to sensitive health information, inaccurate readings, and device malfunctions can compromise the reliability and safety of smartwatch data. Conducting thorough risk assessments helps identify and mitigate potential risks, ensuring the safety, accuracy, and privacy of smartwatch data for users and healthcare providers.

### 4.2. Attacks and Their Impacts

Attacks pose serious risks to the operation, data integrity, and patient safety of these devices. We selected sniffing, jamming, and injection attacks for testing on the above three IoMT devices because they are directly related to the functionality and communication protocols of these devices. By selecting these attacks, we can address potential security vulnerabilities, which could have serious consequences for the user’s privacy.

#### 4.2.1. Sniffing Attack

An oximeter sniffing attack involves intercepting and capturing data transmitted between the oximeter and monitoring systems or devices used by healthcare providers. By intercepting communication channels, attackers can gain unauthorized access to sensitive patient data, such as oxygen saturation levels, heart rate readings, and patient identifiers, leading to inaccuracies in patient monitoring and potentially incorrect clinical decisions. In a smart peak flow meter, attackers gain access to confidential patient data, including peak flow measurements and respiratory rates. Modifying peak flow readings could result in misdiagnosis or inappropriate treatment. In smartwatches, sniffing attacks compromise user privacy by exposing confidential health information, such as heart rate, sleep patterns, and activity levels, to unauthorized parties. Privacy breaches, identity theft, and other malicious activities can occur through unauthorized access to sensitive medical data obtained through sniffing attacks, weakening patient trust in healthcare systems.

#### 4.2.2. Jamming Attack

A jamming attack disrupts the wireless communication between these devices and monitoring systems by interfering with radio frequency signals. It can result in a temporary or prolonged loss of data connectivity, preventing real-time monitoring. It can also delay timely medical interventions for critical care patients, which can cause healthcare providers to miss significant changes in a patient’s condition, increasing the risk of adverse outcomes or complications. It jeopardizes patient safety by impeding the delivery of timely medical care and interventions. Surgical patients and those with respiratory conditions who use oximeters and smart peak flow meters for continuous monitoring may be particularly prone to jamming attacks, which may compromise their treatment and increase the risk of adverse reactions.

#### 4.2.3. Injection Attack

An injection attack involves inserting malicious or unauthorized data packets into the communication stream between these IoMT devices and a monitoring system. The injection of false or misleading information, such as fabricated oxygen saturation readings or alarm signals, can lead to unnecessary clinical interventions. Similarly, injecting misleading data of peak flow measurements can result in incorrect patient assessments. Sometimes, injection attacks may be used to deliver malware payloads or malicious commands to these devices or connected systems. As a result of malware infections, medical devices can be compromised, resulting in system downtime, data breaches, or unauthorized access to sensitive healthcare data. Users relying on smartwatches for health monitoring may be vulnerable to the effects of jamming attacks.

### 4.3. Attack Analysis and Findings

To test the security features during the Bluetooth pairing process of these devices, we implemented sniffing, jamming, and injection attacks where we captured the traffic sent between devices. We conducted a BLE attack against the above three devices while connecting them through their mobile applications. It allowed us to listen to only BLE devices and capture their traffic.

We used Btlejack and Mirage [37] as a tool for sniffing, jamming, and injecting BLE devices. It is primarily used to test the security of devices that use Bluetooth as a communication protocol. Ble_hijack implements active attacks allowing the hijack of either the slave or master role of a BLE connection. It can perform a jamming-based Btlejack attack, allowing the hijacking of both master and slave roles and the hijacking of a new connection or an established connection.

#### Test Bed

A test bed for analysing network traffic was created using Wireshark (version 3.6.18). To carry out the sniffing attack, we captured the BLE communication between an Android phone and an oximeter and a smartwatch. When Bluetooth devices are connected to each other, they are called master–slave relationships, where one device is the master device, and the other one is the slave. The master device sends information to the slave device, and the slave device listens to the master information.

Figure 7 and Figure 8 are divided into three parts: the packet list pane, packet detail pane, and packet byte pane. They work together to provide a detailed view of a captured packet. The packet detail pane offers a high-level, human-readable interpretation of the packet structure and content, whereas the packet byte pane allows for a more granular, in-depth examination of the raw data underlying the communication.

In the packet list panes in Figure 7 and Figure 8, the leftmost column shows numbered entries representing individual data packets exchanged between the phone and the oximeter, followed by the time each packet was captured. It shows BLE communication between an oximeter (Slave_exaf9ab4dd) and an Android phone, which is the master device. The protocol used here is BLE, which is useful for wearable medical devices and sensors because it reduces power consumption and memory requirements. Basically, it is designed to operate in sleep mode and wake up only when a connection is initiated. This improves efficiency when discovering devices and during connection procedures and results in packets with shorter lengths, while services and protocols are simpler. We can see it sends out a huge number of empty PDUs to jam the communication in a short period of time.

Data packets are sent from the master device to the oximeter, which replies with an “Empty PDU” packet. In two instances, the oximeter sends a “Rcvd Handle Value Notification” message after receiving a packet from the master device, highlighting the successful attack.

Expanding the Bluetooth Attribute Protocol section of a package (as shown in the image below) provides the following information:The type of operation performed (*read/write/notify*);The characteristic on which it was carried out;The transmitted data.

Packet number 3081 has been selected, where the packet bytes pane shows the data of the current packet (selected in the “Packet List” pane) in a hex dump style. Each line contains the data offset, sixteen hexadecimal bytes, and sixteen ASCII bytes. A **“hex dump”** represents a binary data stream where the contents of that stream are displayed as hexadecimal values. It divides the binary data into 8-bit bytes and displays the value of each byte as a two-digit hexadecimal number. The packet bytes pane displays a hex dump of the corresponding packet data. Opcode here specifies the action being performed in a particular packet. Handles are used to identify specific data characteristics on a device. Value refers to the actual data payload being exchanged between the devices.

Similarly, Figure 8 shows the data packets exchanged between the phone and smartwatch, highlighting the attack as being successful.

Furthermore, a jamming attack was conducted using microbit and Mirage, as shown in Figure 9. ble_jam allows the use of the jamming features implemented in BTLEJack and allows the jamming of a new connection or an existing connection in JAMMING_MODE. Here, the value of the input parameter is set to “existing connection”. Access address, CRCInit, and channel map are provided as additional parameters for targeting a specific device.

Here, we tried to exploit a risk in a program called “Mirage” by overflowing a buffer with data. This overflow corrupts the program’s memory and allows us to inject and execute malicious code. We initiated the attack by sending a specially crafted message to the target program. The message overflows a buffer in the program’s memory, corrupting it and allowing us to inject the code. The injected code reads the program’s memory to recover various configuration parameters, including the channel map, hop interval, and hop increment. Once all the parameters were recovered, we tried to establish a connection using the stolen parameters where the connection was lost, but we successfully retrieved all the configuration data.

This test bed setup enables the comprehensive testing and validation of IoMT solutions to ensure their reliability and effectiveness in IoMT devices.

### 4.4. HRA Analysis

Using data from the case study conducted, the risk level is calculated. The input parameters are the frequency of occurrence and consequence severity of the risk events. Based on Table 5, we create a pair-wise comparison matrix with the help of the scale of relative importance. We can replace the crisp numeric values with fuzzy numbers and similarly convert their reciprocal values into fuzzy numbers. The reciprocal fuzzy number ${\tilde{A}}^{-1}$ is computed using the equation ${\tilde{A}}^{-1}={\left(l,m,u\right)}^{-1}=\left(\frac{1}{u},\frac{1}{m},\frac{1}{l}\right)$. Here, the crisp numeric value 5 for strongly important has the fuzzy number (4,5,6); therefore, its reciprocal is ${\tilde{A}}^{-1}={\left(4,5,6\right)}^{-1}=\left(\frac{1}{6},\frac{1}{5},\frac{1}{4}\right)$.

Likewise, all the values are converted into fuzzy and reciprocal fuzzy values to obtain the fuzzified pair-wise comparison matrix given in Table 6.

Next, we calculate the weight using the geometric mean method in Table 7, which gives us the fuzzy weight for each criterion using Equation (3). These fuzzy weights are then defuzzified to obtain crisp numeric values. The defuzzification method used here is the Centre of Area (COA). The overall RL is calculated using Equation (6) for all three IoMT devices. For example, it can be seen that the injection attack on the smartwatch has the highest risk level of 8.57, which is of “Extreme importance” and must be treated first. Considering all the risk outcomes, the device should be treated where the risk level is highest. The detailed calculations and explanations are provided in Appendix A.

Obtaining fuzzy weights helps prioritize criteria and alternatives in decision-making. It allows us to evaluate the relative importance of different factors and make informed decisions based on these comparisons.

Based on these insights, the case study supports that the suggested HRA approach can be applied in everyday IoMT devices and can be applied for their risk assessment with expert knowledge. It can provide insights into the potential uncertainties of the assessment process. However, there are certain limitations of this paper, which will be considered in future work:In this research, we considered only research papers for the literature review, excluding conference papers, review papers, book chapters, and non-English papers.Our study is focused on the applicability of fuzzy logic and the FAHP-based approach, while other approaches might be possible for risk assessment.

## 5. Conclusions and Future Work

IoMT devices have become increasingly popular for monitoring heart rate, lung function, exercise, and sleep patterns. However, the increasing popularity of these devices also raises concerns about data security. Manufacturers must ensure the confidentiality, security, and accessibility of the data collected. This facilitates accurate health tracking, fosters user trust, and prompts timely medical consultations. As these technologies evolve and incorporate more sensors, the risk of attackers obtaining sensitive real-time data and profiling potential victims increases.

We present an advancement in the field of risk assessment for IoMT devices. Our proposed approach, utilizing fuzzy logic and the FAHP, offers a practical solution for determining risk levels (RLs). This approach is demonstrated through a case study involving an oximeter, smartwatch, and smart peak flow meter. The potential of our proposed method for the risk assessment of IoMT devices is notably effective when risk data are incomplete or a high level of uncertainty is involved. By incorporating fuzzy logic and the FAHP, this approach can effectively leverage domain experts’ experience and risk management knowledge. It can also transform information from various sources into a knowledge base, including qualitative descriptors, MFs, and fuzzy rules used in the fuzzy inference process. Our study demonstrates that risk analysis based on fuzzy logic and the FAHP approach provides a reliable tool for risk analysis in diverse circumstances. The outcome will be beneficial for demonstrating policy adherence to cybersecurity recommendations for everyday-use IoMT devices.

Our study examines security concerns for three specific IoMT devices, providing some insights into the risks. However, the broader IoMT ecosystem encompasses a diverse range of devices with varying vulnerabilities, requiring further research. In order to address this, future research will expand testing to include small to medium-sized IoMT devices (such as portable vital monitor or home-based health monitoring system) and scale up to explore security issues in medium to large devices (such as imaging systems).

Our analysis can be helpful for manufacturers who design these devices. Considering the rapid and unstoppable integration of multiple technologies into the medical field, more IoMT devices are expected to be adopted by people. We have not found much research evaluating medical devices used by users on a daily basis, so our findings could be used for experiments on medical IoT devices, like an oximeter, smartwatch, and smart peak flow meter.

## Figures and Tables

**Figure 1 sensors-24-03223-f001:**
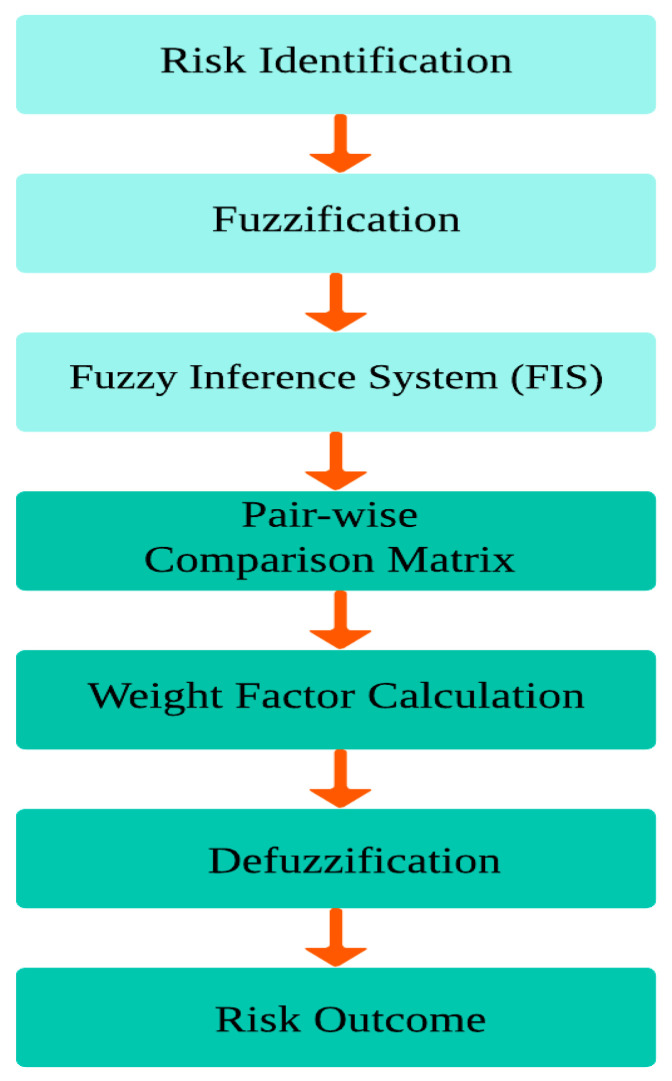
Flowchart of hybrid risk assessment.

**Figure 2 sensors-24-03223-f002:**
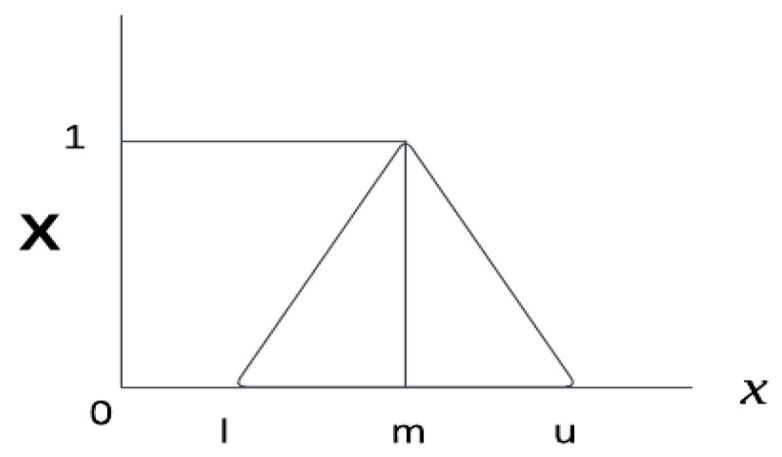
Triangular fuzzy number [26].

**Figure 3 sensors-24-03223-f003:**
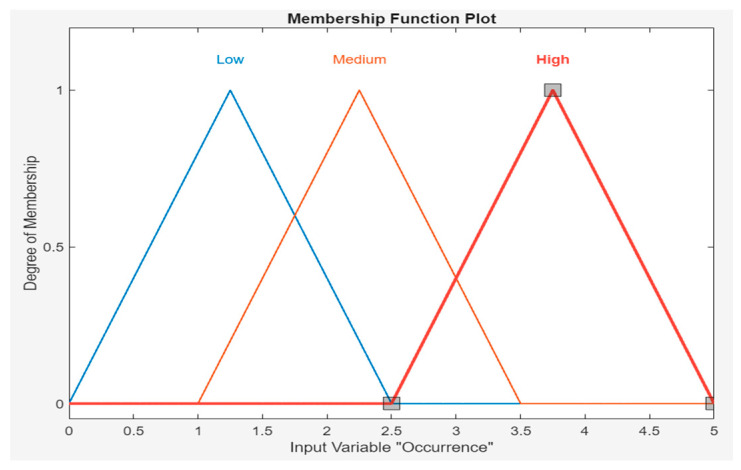
MF of frequency of occurrence.

**Figure 4 sensors-24-03223-f004:**
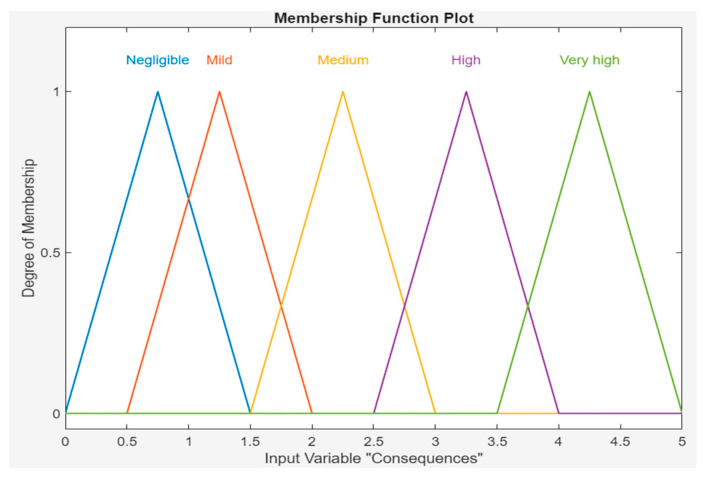
MF of severity of consequences.

**Figure 5 sensors-24-03223-f005:**
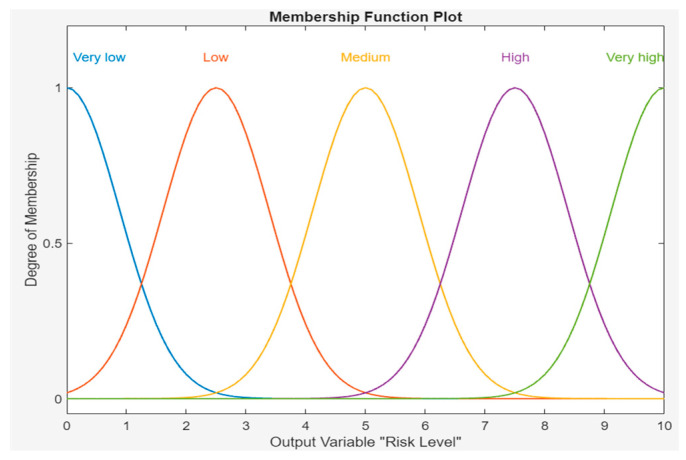
MF of risk level.

**Figure 6 sensors-24-03223-f006:**
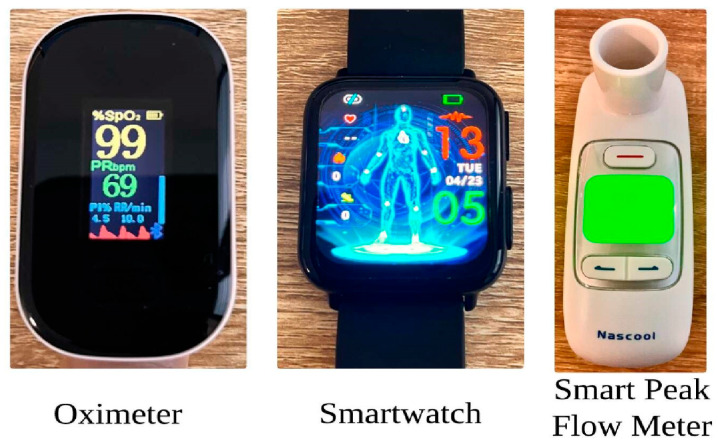
Devices used for testing [33,34,35].

**Figure 7 sensors-24-03223-f007:**
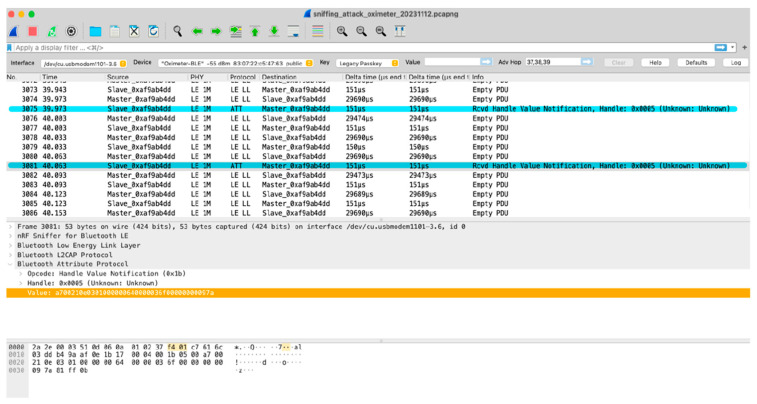
Sniffing attack on oximeter.

**Figure 8 sensors-24-03223-f008:**
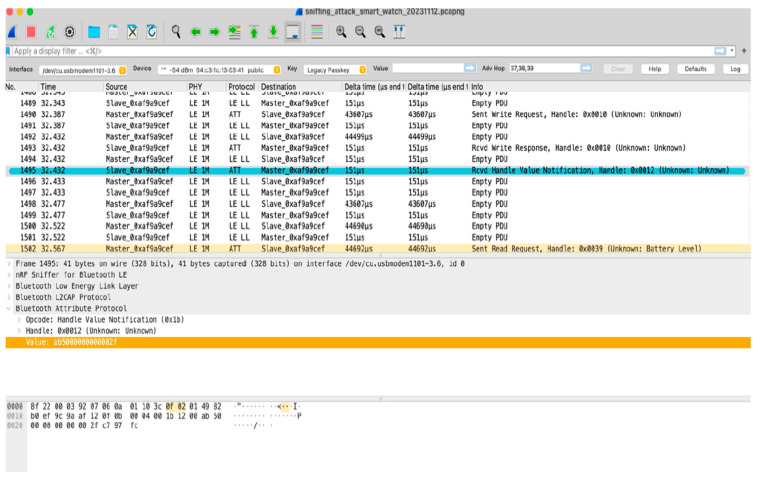
Sniffing attack on smartwatch.

**Figure 9 sensors-24-03223-f009:**
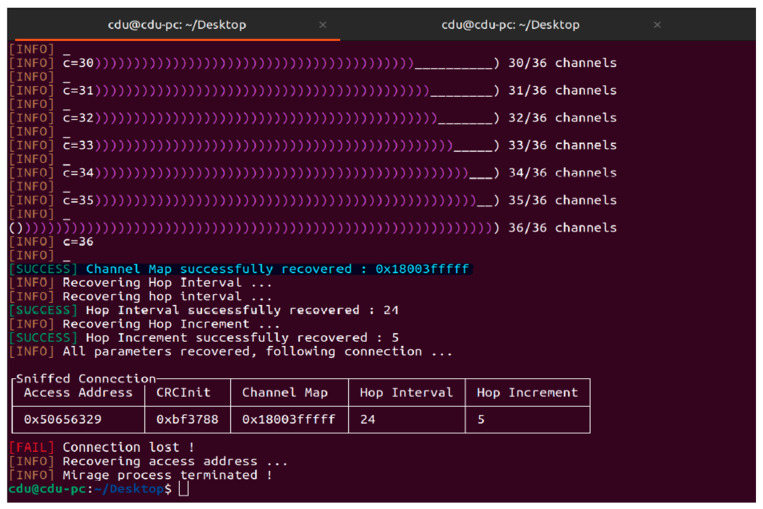
Jamming attack on oximeter.

**Table 1 sensors-24-03223-t001:** Frequency of occurrence.

Qualitative Expression	Description	Triangular Parameters
Low	Unlikely to occur due to strong security measure	[0 1.25 2.5]
Medium	Expected to occur due to occasional lapses in security	[1 2.25 3.5]
High	Highly expected to occur due to no security in place	[2.5 3.75 5]

**Table 2 sensors-24-03223-t002:** Severity of consequences.

Qualitative Expression	Description	Triangular Parameters
Negligible	Minor disruption, minimal impact on device	[0 0.75 1.5]
Mild	Some disruption in functionality, does not compromise patient safety	[0.5 1.25 2]
Medium	Moderate disruption, recovery may require moderate effort	[1.5 2.25 3]
High	Severe disruption leading to compromised patient safety	[2.5 3.25 4]
Very high	Critical disruption posing a serious risk to patient safety	[3.5 4.25 5]

**Table 3 sensors-24-03223-t003:** Risk levels.

Qualitative Expression	Description	Gaussian Parameters
Very low	Acceptable risk	[0.8847 2.776 × 10^−17^]
Low	Tolerable risk	[0.8847 2.5]
Medium	Reduced risk with reasonable controls	[0.8847 5]
High	Unacceptable risk but may be reduced with controls	[0.8847 7.5]
Very high	Unacceptably high risk	[0.8847 10]

**Table 4 sensors-24-03223-t004:** Fuzzy Rules.

Rule	Explanation
R1	If occurrence is low and consequences are negligible, then risk level is very low
R2	If occurrence is medium and consequences are negligible, then risk level is low
R3	If occurrence is high and consequences are negligible, then risk level is medium
R4	If occurrence is low and consequences are mild, then risk level is low
R5	If occurrence is medium and consequences are mild, then risk level is low
R6	If occurrence is high and consequences are mild, then risk level is medium
R7	If occurrence is low and consequences are medium, then risk level is medium
R8	If occurrence is medium and consequences are medium, then risk level is medium
R9	If occurrence is high and consequences are medium, then risk level is high
R10	If occurrence is low and consequences are high, then risk level is medium
R11	If occurrence is medium and consequences are high, then risk level is high
R12	If occurrence is high and consequences are high, then risk level is high
R13	If occurrence is low and consequences are very high, then risk level is high
R14	If occurrence is medium and consequences are very high, then risk level is very high
R15	If occurrence is high and consequences are very high, then risk level is very high

**Table 5 sensors-24-03223-t005:** Risk level representation for FAHP.

Linguistic Term	Crisp Numeric Value	Triangular Fuzzy Scale	Reciprocal Fuzzy Scale
Equally important	1	(1,1,1)	(1,1,1)
Intermediate value	2	(1,2,3)	(1/3,1/2,1)
Moderately important	3	(2,3,4)	(1/4,1/3,1/2)
Intermediate value	4	(3,4,5)	(1/5,1/4,1/3)
Strongly important	5	(4,5,6)	(1/6,1/5,1/4)
Intermediate value	6	(5,6,7)	(1/7,1/6,1/5)
Very strongly important	7	(6,7,8)	(1/8,1/7/1/6)
Intermediate value	8	(7,8,9)	(1/9,1/8,1/7)
Extremely important	9	(8,9,9)	(1/9,1/9,1/8)

**Table 6 sensors-24-03223-t006:** Fuzzified pair-wise comparison.

Device	Attack Type	Sniffing	Jamming	Injection
Oximeter	Sniffing	(1,1,1)	(4,5,6)	$\left(\frac{1}{5},\frac{1}{4},\frac{1}{3}\right)$
	Jamming	$\left(\frac{1}{6},\frac{1}{5},\frac{1}{4}\right)$	(1,1,1)	$\left(\frac{1}{4},\frac{1}{3},\frac{1}{2}\right)$
	Injection	(3,4,5)	(2,3,4)	(1,1,1)
Smartwatch	Sniffing	(1,1,1)	(5,6,7)	$\left(\frac{1}{8},\frac{1}{7},\frac{1}{6}\right)$
	Jamming	$\left(\frac{1}{7},\frac{1}{6},\frac{1}{5}\right)$	(1,1,1)	$\left(\frac{1}{5},\frac{1}{4},\frac{1}{3}\right)$
	Injection	(6,7,8)	(3,4,5)	(1,1,1)
Smart peak flow meter	Sniffing	(1,1,1)	(3,4,5)	(2,3,4)
	Jamming	$\left(\frac{1}{5},\frac{1}{4},\frac{1}{3}\right)$	(1,1,1)	(5,6,7)
	Injection	$\left(\frac{1}{4},\frac{1}{3},\frac{1}{2}\right)$	$\left(\frac{1}{7},\frac{1}{6},\frac{1}{5}\right)$	(1,1,1)

**Table 7 sensors-24-03223-t007:** Overall weights and risk levels.

Device	Attack Type	$\mathrm{Fuzzy}\;\mathrm{Weight}\;{\tilde{w}}_{i}$	$\mathrm{Weight}=w_{i}$	Risk Level
Oximeter	Sniffing	(0.207, 0.2857, 0.4069)	0.2998	2.998
	Jamming	(0.0762, 0.10717, 0.1620)	0.11512	1.04
	Injection	(0.4065, 0.6071, 0.8794)	0.631	5.048
Smartwatch	Sniffing	(0.1752, 0.2192, 0.2784)	0.2242	3.14
	Jamming	(0.0626, 0.08009, 0.1071)	0.0832	1
	Injection	(0.5372, 0.7008, 0.9044)	0.7141	8.57
Smart Peak Flow Meter	Sniffing	(0.4033, 0.60002, 0.8633)	0.6222	4.97
	Jamming	(0.2220, 0.30001, 0.4217)	0.3145	3.46
	Injection	(0.0726, 0.0999, 0.1476)	0.1067	1.07

## Data Availability

The original contributions presented in the study are included in the article, further inquiries can be directed to the corresponding authors.

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
