# Peer review of "Risk Evaluation and Attack Detection in Heterogeneous IoMT Devices Using Hybrid Fuzzy Logic Analytical Approach"

_sensors, 2024, doi:10.3390/s24103223_

Round 1

Reviewer 1 Report

Comments and Suggestions for Authors

The present work deals with the cyber risk assessment in IoT environments of medical systems. The authors present an approach based on fuzzy logic and Fuzzy Analytical Hierarchy Process (FAHP) to evaluate the individual and overall risks of threats in such environment.

The method was validated in a setting where three types of attacks were performed: Bluetooth Low Energy (BLE) attacks (sniffing, jamming and injection) on oximeter, smart watch, and smart peak flow meter devices.

The method and process are not new in the literature and the authors provide adequate references. The novelty comes in form of the prove of applicability of the method to healthcare IoT devices.

The major concern raised during the review is that the rationale for the selection of the Triangular function as MF for both occurrences and consequences seems to be quite arbitrary for "better understanding", when it should be either based on actual data of humans assessing such factors (historical data) or human experts' opinion about the typical shape that such factors take during the assessments (which are usually uncertainties-prone human assessments, and this is why fuzzy logic is employed to explain such uncertainties).

Therefore, the conclusions that smartwatches were found to have a risk level of 8.57 for injection attacks, while jamming attacks had the lowest risk level of 1, may not be based on reality, but be only an example assessment.

I recommend the authors to please explain the rationale of MF function selection.

The explanation of the attacks is clear and it will be welcome by non-expert readers.

The paper is well written and well structured. It is easy to read and the English language used is fine, no issues were detected.

Reviewer 2 Report

Comments and Suggestions for Authors

The paper presents a relatively simple analysis using well-known tools and security frameworks on commercial devices. Nevertheless, the presentation of both the theoretical part and the experimental results was good, and I did not identify significant issues in the analysis.

Some points that should be addressed before the eventual publication of this work are the following:
- In the abstract, it is not expressed what the risk level represents. At this point, it just seems an arbitrary numeric value.
- There is an issue with the reference format. Sometimes, they are reported as superscripts, and other times, they are between square brackets.
- It is unclear how the parameters have been chosen for the membership function in Section 3.1.3.
- The work used some open-source tools to assess the security of only a limited number of IoMT devices. While the paper provides some empirical evidence about the security concerns about these specific devices, it is unclear whether such issues impact a larger set of devices.

Comments on the Quality of English Language

The quality of English is good enough for possible publication.
